# Comparison of Two Different Sedation Protocols during Transvaginal Oocyte Retrieval: Effects on Propofol Consumption and IVF Outcome: A Prospective Cohort Study

**DOI:** 10.3390/jcm10050963

**Published:** 2021-03-01

**Authors:** Paraskevi Matsota, Tatiana Sidiropoulou, Tereza Vrantza, Maria Boutsikou, Elena Midvighi, Charalampos Siristatidis

**Affiliations:** 1Second Department of Anesthesiology, “Attikon Hospital”, Medical School, National and Kapodistrian University of Athens, Rimini 1, Chaidari, 12642 Athens, Greece; pmatsota@med.uoa.gr (P.M.); tsidirop@med.uoa.gr (T.S.); elenimidvighi@gmail.com (E.M.); 2Assisted Reproduction Unit, Third Department of Obstetrics and Gynecology, “Attikon Hospital”, Medical School, National and Kapodistrian University of Athens, Rimini 1, Chaidari, 12642 Athens, Greece; vrantzat@hotmail.com; 3Cardiology Department, Mediterraneo Hospital, 16675 Athens, Greece; boutsikoum@gmail.com; 4Adult Congenital Heart Disease Unit/MRI Unit, Royal Brompton Hospital, London SW3 6NP, UK; 5Assisted Reproduction Unit, Second Department of Obstetrics and Gynecology, “Aretaieion Hospital”, Medical School, National and Kapodistrian University of Athens, Vas. Sofias 76, 11528 Athens, Greece

**Keywords:** conscious sedation, dexmedetomidine, propofol, in vitro fertilization (IVF), pregnancy rates

## Abstract

(1) Background: There has been various reports on the potential impact of anesthetic agents used during oocyte retrieval (OR) on the impairment of the capacity of the oocyte for fertilization and subsequent embryo quality; results have been conflicting; (2) Methods: The effects of two different sedation protocols during OR in two groups of patients undergoing In Vitro Fertilization/Intra-Cytoplasmic Sperm Injection IVF/ICSI, were compared on propofol consumption and on in vitro fertilization (IVF)/ICSI success. The study group received dexmedetomidine and fentanyl, while the control remifentanil and midazolam. In a prospective cohort study, we encompassed 72 cycles/patients. The administered dose of propofol per patient and fertilization rates were the primary outcomes, while anesthesiological parameters and IVF/ICSI outcomes were the secondary endpoints; (3) Results: We found a significant increase in propofol consumption in the study compared to the control group (77.0 ± 10.6 mg vs. 12.1 ± 6.1; *p* < 0.001), but fertilization rates were similar (*p* = 0.469). From the secondary anesthesiological outcomes, the post anesthesia discharge scores were better in the control group (15.0 (13.5 min) vs. 5.0 (10.0 min), *p* = 0.028). From the IVF/ICSI secondary outcome parameters, we found a higher quality of embryos on day three in the study compared to the control group (*p* = 0.040). The comparison of the other secondary outcomes yielded non-significant differences; (4) Conclusions: The use of dexmedetomidine, as an alternative agent during OR, was associated with higher propofol consumption as a rescue dose compared to remifentanil but was linked with similar fertilization rates and higher quality of embryos produced.

## 1. Introduction

Oocyte retrieval (OR) is a short surgical procedure, which is usually performed under general anesthesia, sedation or Monitored Anesthesia Care (MAC), although local or regional anesthesia can be altenative methods [1]. There is data that shows that conscious sedation (CS)—when used in OR—compared to general anesthesia is associated with higher in vitro fertilization (IVF) success rates, in terms of higher pregnancy and live birth rates [2]; however, the optimal/effective analgesic-sedative regimen has not been determined yet [3]. In this context, a fast-track anesthetic regimen, consisting of a combination of anesthetic, sedative and analgesic drugs with the appropriate pharmacokinetic profile, in this day-case operation, would be ideal.

Three are the most frequently used drugs for sedation during OR: propofol, the intravenous anesthetic of choice, due to its rapid onset and short duration of action; midazolam, the shortest-acting available benzodiazepine; and remifentanil, an ultrashort-acting synthetic opioid [4,5,6]. Dexmedetomidine (DEX) is a highly selective α2 adrenergic receptor agonist also used to provide CS in both surgical and non-surgical interventions, due to its sedative and analgesic properties [7,8,9]. There is strong evidence that its use, when compared to midazolam combined with paracervical block in OR, provides better post-procedural analgesia and higher overall patient satisfaction [10]. Its main advantage is the lack of respiratory depression. In animal models, it has been linked with positive effects at the level of receptors, of both neurons and endothelium [11,12].

There has been high scientific interest on the potential impact of anesthetic agents used in OR on oocyte fertilization and embryo quality. Drugs, such as propofol, thiopental, midazolam, fentanyl and alfentanil, along with local anesthetics, can accumulate in the follicular fluid and may affect oocyte structure and subsequent fetal development [13,14,15,16,17]. The assessment of their potential role on IVF success has yielded conflicting results. A review encompassing 43 clinical trials, showed that both alfentanil and remifentanil are not associated with toxic effects, but the role of propofol remains controversial [18]. Moreover, experimental studies have revealed a potential negative impact of propofol on oocytes. In specific, in mice, propofol exposure has been acquainted to exert a toxic effect on the ability of oocytes to be fertilized in a dose- and time-dependent manner [19,20]. Hence, concerns have been raised on the suitability of propofol as an anesthetic drug for OR [21]. In this context, it could be assumed that a low dose during sedation in OR would exert a lower impact, in terms of toxicity, on oocyte maturation.

The aim of this study was to compare the effect of two different regimens for conscious sedation during OR, namely MAC with dexmedetomidine and fentanyl versus MAC with midazolam and remifentanil, on total propofol administration (given as intermittent IV rescue sedative dose) and IVF outcome, concerning fertilization rates.

## 2. Materials and Methods

This is a single-center two-arm prospective cohort study conducted at the Assisted Reproductive Unit of the Third Department of Obstetrics and Gynaecology in collaboration with the Second Department of Anesthesiology at the “Attikon” University Hospital (Athens, Greece). The study protocol was approved by the Hospital Ethics and Research Committee (AN 27/7/2017) and registered in clinicaltrials.gov (NCT03362021). The study was performed in accordance with good clinical practice guidelines and the ethical standards of the Helsinki Declaration, lasted from November 2017 to November 2019, while written informed consent was obtained from all patients. All OR procedures were performed by the same gynecologist.

### 2.1. Patient Population/Eligibly Criteria

Women scheduled for OR undergoing IVF/ICSI under CS were enrolled for the present study. Following clinical evaluation along with previous medical and reproductive history, patients’ demographic parameters were recorded along with the respective hormonal profiles and treatment protocols.

Inclusion criteria for the study entry were: infertile women with an indication for IVF/ICSI; age 25–43 years; cycle length between 22 and 35 days; and basal FSH ≤ 11 mIU/mL.

Exclusion criteria, in terms of infertility history, included: history of more than three previous unsuccessful IVF/ICSI cycles; woman’s age > 43 years; levels of basal FSH > 12 IU/L; previous poor ovarian response; ovarian surgery and pathology affecting the endometrial cavity, hereditary or acquired thrombophilia along with surgically proven endometriosis. Criteria in terms of anesthesia were: patient refusal to participate; personal history of allergy to any drug used; epilepsy; gastroesophageal reflux; morbid obesity; history of severe cardiovascular or other systemic disease (ASA > 2), severe hypotension or bradycardia; and presence of atrioventricular block or stroke.

### 2.2. Groups, Anesthesia Management and Assessments

Patients were fasted and unpremedicated and consecutively allocated into one of the two groups:

Group DEX: patients received dexmedetomidine (solution 4 mcg/mL) through continuous infusion at a dose of 1 mcg/kg/min, starting 10 min before the OR until the end of the procedure and fentanyl 100 mcg iv.

Group REMI: patients received remifentanil (solution 50 mcg/mL) through continuous infusion at a dose of 0.2 mcg/kg/min, starting 10 min before the OR until the end of the procedure and midazolam 1 mg iv.

The allocation of patients was achieved using a 1:1 proportional pattern, depending on the random order in which they entered the Unit. Notably, both patients and gynecologist were not aware of the study protocol used.

In both groups, in cases of non-co-operation (e.g., kinetic response), propofol was administered intravenously as a bolus sedative rescue dose, as follows: 1mg/kg as the first dose and 0.5 mg/kg for the next ones. At the end of the procedure, all doses were recorded.

Ringer’s lactate solution was infused during and immediately after the OR. During the OR, patients were spontaneously breathing via a Venturi mask providing a 50% oxygen-enriched air. The intraoperative monitoring included: ECG, noninvasive blood pressure (NIBP), SpO_2_, EtCO_2_ and depth of sedation (BIS, OAA/S). Adverse effects included: hypotension (defined as systolic arterial pressure [SAP] < 20% from baseline or SAP < 80 mmHg), bradycardia (defined as heart rate [HR] < 20% from baseline or HR < 50 b/min), rigidity, airway obstruction, and need for bag-mask ventilation (whenever abolition of spontaneous ventilation or decrease of SpO_2_ < 90% was observed). In cases of bradycardia or hypotension, atropine 0.5 mg or ephedrine 5 mg i.v. were administered, respectively.

At the end of the OR, paracetamol 1 gr i.v. was administered. Postoperative monitoring included ECG, NIBP and SpO_2_ every 15 min, along with pain (VAS scores) and nausea/vomiting (PONV) evaluation. The quality of recovery was assessed using the time required to achieve the maximum OAA/Score, along with the time required to achieve a PADSS score ≥9. Prior to the patient’s discharge from the Unit, both patients’ and gynecologist’s overall satisfaction scores, related to the sedation technique, were also assessed.

Scales and scores used during monitoring are presented in Table 1.

### 2.3. IVF Protocol

IVF protocols used were (i): long protocol with GnRH—agonists and (ii) short protocol with GnRH—antagonists. Ovarian stimulation was achieved with rFSH, and described in our previous studies [22,23]. OR was scheduled 36 h after oocyte triggering and when there were at least 3 follicles >17 mm in diameter. ICSI was performed in all cases to ensure fertilization. Vitrification was the method of freezing of supernumerary embryos; embryo transfers were conducted at day three, while the maximum number of embryos transferred was two, in accordance with the Hellenic legislation.

### 2.4. Outcome Measures

Primary endpoints were: the administered dose of propofol per patient and fertilization rates.

Secondary endpoints were:(1)Anesthesiological parameters: total dose of dexmedetomidine or remifentanil administered, hemodynamic parameters, dose of ephedrine, dose of atropine, BIS values, OAA/S score, adverse effects during anesthesia, time required to achieve the maximum OAA/S score, PONV, VAS, time required to achieve PADSS score ≥9 and the overall patients’ and gynecologist’ satisfaction.(2)IVF outcomes (definitions used were as reported in Zegers-Hochschild et al., 2017 [24]): number of oocytes retrieved at the day of the OR, number of fertilized embryos [number of oocytes with two nuclei (2PN) divided by the total number of oocytes retrieved], embryo quality and number of top quality embryos at day 3 (Veeck: 5-point scale: Grade 1 = excellent, Grade 2 = good, 3 = moderate, 4 = poor, 5 = unsustainable), cycle cancelation (in cases with premature ovulation, no oocytes retrieved or no embryos available for transfer), biochemical (defined as a positive pregnancy test), ectopic, clinical pregnancy (defined as the presence of fetal heart beat at 7 weeks gestation) and miscarriage rates (defined as pregnancy loss up to the 20th week of pregnancy per positive pregnancy test). Embryo quality was assessed according to morphological criteria based on the overall blastomere number, size, appearance and degree of fragmentation [25].

### 2.5. Sample Size

Using the G*Power 3.0.10 software, we found that a sample size of 86 patients in each group would be necessary to detect a 30% dose difference in propofol using Fisher’s exact double-sided alpha test at the 0.05 level, including a 5% withdrawal rate from the study (Appendix A). Unfortunately, the study was prematurely ended as one of the main investigators moved to another IVF Unit, and because of the Covid-19 pandemic: IVF protocols in all Units worldwide changed from fresh to frozen, so that it was impossible to recruit more patients during the last year.

A post hoc power analysis of the data on propofol consumption was also carried out to check the power of the study results, in an effort to reduce the bias associated with the premature ending of the study.

### 2.6. Statistical Analysis

Normality of the data was examined by Kolmogorov-Smirnov test. Data are expressed as mean ± standard deviation for normally distributed variables or median (IQR) otherwise. The variable regarding propofol dose was extremely skewed, thus the data reading this variable was presented as trimmed mean± SE. Independent samples Student t-test was applied to detect differences between groups, where continuous variables were normally distributed. Otherwise, Mann–Whitney U test was applied. Categorical variables were presented as Number (%). Chi square (X^2^) was used to detect differences between categorical variables. Parametric tests were applied in order to detect differences in the percentage of the number of oocytes with two nuclei (2PN) divided by the number of oocytes recovered during OR per patient, since the assumptions of the homogeneity of Variances (Levene test) and the equality of means (Welch test) were satisfied. *p* < 0.05 was considered statistically significant. We used the SPSS software (version 23; SPSS Inc, Chicago, IL, USA).

## 3. Results

### 3.1. Study Characteristics

A total of 72 cycles, stemming from a total of 72 infertile couples, comprised the cohort in our study. The initial cohort were 77 patients; of them, five were excluded: two in the DEX group, because of refusal to participate and due to morbid obesity, respectively; and three in the REMI group: one due to gastro esophageal reflux and two due to morbid obesity, respectively. Demographic data were similar between groups (Table 2).

### 3.2. Primary Outcomes

There was a significant increase in propofol consumption in group DEX compared to group REMI (77.0 ± 10.6 mg vs. 12.1 ± 6.1; *p* < 0.001; Table 3a), but fertilization rates were similar (*p* = 0.469; Table 3b).

### 3.3. Secondary Outcomes

Regarding the secondary outcomes: all anesthesiological parameters did not differ among groups (*p*-values > 0.05), apart from the post anesthesia discharge score, where patients could be discharged earlier in the REMI group compared to the DEX group (*p* = 0.028; Table 3a; Figure 1). IVF outcome parameters were similar between groups, except from a higher quality of embryos on day three found in group DEX compared to group REMI (*p* = 0.040; Table 3b). We also observed a marginal difference between groups in the number of top-quality embryos at day three (*p* = 0.052) and in miscarriage rates (*p* = 0.051) (Table 3b).

A post hoc power analysis of the data on propofol consumption, showed a power of 99% (two-tailed; effect size d = 1.24; α err prob = 0.05; sample size group 1 and 2 = 36 each; noncentrality parameter = 5.29; critical t = 1.99; Df = 70).

## 4. Discussion

The purpose of this single center cohort study was to compare the effects of two different sedation protocols during transvaginal oocyte retrieval (OR) on propofol consumption as a rescue sedative and on IVF success, in terms of fertilization rates. The study group received dexmedetomidine (DEX) and fentanyl, while the control remifentanil and midazolam. We found higher consumption of propofol in the study compared to the control group, but similar fertilization rates. As for the secondary outcomes preset for this study, only the quality of embryos three days post OR were higher, while discharge scores were lower in the study compared to the control group. In the present study, we found that DEX was associated with increased requirements of rescue propofol during OR. To the best of our knowledge, there is only one similar study investigating the use of DEX in the clinical setting of OR, which contradicts our findings, due to its different methodology and doses of the drugs administered [10]: authors compared DEX and midazolam in women undergoing OR under conscious sedation combined with paracervical block (PCB) and found that DEX, compared to midazolam, was associated with decreased demand for rescue propofol administration. It is worth noting that the study revealed the superiority of the sedative effect of DEX vs. midazolam, as the procedural pain was adequately controlled with the PCB [10]. In our study, it seems that remifentanil administration achieved better procedural pain control, resulting in significantly less propofol consumption. However, the increased total consumption of propofol observed in the study group did not increase the need for assisted ventilation during the OR nor delayed the immediate recovery of patients, as there was no difference between the two groups in terms of the time required to achieve the maximum OAA/S score. Moreover, no differences were found in the rest of the secondary anesthesiological outcomes, including hemodynamic parameters, pain, PONV and adverse effects, except from the PADSS scores, which were better in the control group, indicating that patients could be discharged earlier. There was no episode of severe hypotension or bradycardia requiring medication, while most patients were free of postoperative pain and PONV, and expressed satisfaction with the sedative technique used. In contrast, the observed statistical difference with respect to the discharge scores is of questionable clinical significance, as the mean delay time in patients receiving DEX was approximately only 10 min longer. This is also supported by the fact that the gynecologist, who was blind to the patient’s group, expressed similar satisfaction with both sedative regimens applied during OR. This leads to our conclusion that, from an anesthesiological point of view, both techniques of conscious sedation are efficient when used in OR.

Concerning IVF outcomes, we observed similar fertilization rates, despite the fact that higher doses of propofol have been administered as a rescue dose in the study group. Thus, we have not observed a negative impact of propofol on oocyte quality and fertilization when combined with DEX, even though initial and experimental studies have postulated that it may negatively affect the oocytes [2,19,21,26]. One possible explanation for this discrepancy may be the fact that even higher doses of propofol used in the DEX group were significantly lower than those used in studies related to the negative effect of propofol on the reproductive outcome [27,28]. Specifically, in these studies propofol was administered as a bolus dose of 2.5 mg/kg followed by continuous infusion of 200 microgram/kg/min [27] or 500 microgram/kg/min [28], respectively, and was associated with a significantly higher rate of abnormal fertilization. Our hypothesis is also supported by experimental data showing that the toxic effect of propofol on the ability of oocytes to be fertilized is dose-dependent [19,20].

Another possible explanation could be a possible protective effect of DEX on reproductive outcome. It is worth mentioning that a better overall quality of embryos at day three was associated with the use of DEX in our study. This finding could be attributed to a potential protective effect of DEX on the quality of oocytes, given that experimental data are in support with the protective effects of DEX at the level of both neuron receptors and vascular endothelium [11,12]. However, despite the better quality of embryos observed in DEX group, the rates of clinical pregnancy and miscarriage were not significantly different between the groups studied.

Direct comparisons with the literature is rather difficult, as there was only one study on the use of DEX during OR [10], which ended up with similar results to ours, concerning the number of oocytes obtained, embryos transferred and percentage of pregnancies per embryo transferred. Conclusively, our results showed that the use of DEX may overwhelm the potential negative effect of the higher use of propofol, as fertilization rates (preset as a primary outcome) were not affected, along with clinical pregnancy and miscarriage. Moreover, we observed that DEX use was linked with better quality of embryos produced, a finding that could be attributed to a potential protective effect of DEX on the quality of oocytes given that both remifentanil [18] and midazolam [29] have not been associated with toxic effects. This observation remains a postulation, as there are no data from clinical trials to support it, as yet.

### Limitations and Strengths

The strength of the study lies on its prospective design, with a priori set parameters for the evaluation of the two anesthetic protocols. The predetermined dataset was extensive and detailed, representing all clinical steps during the application of anesthesia in OR. Limitations of the study include its non-randomized design that is linked with known and unknown confounding and biases [30] and the small cohort size. The latter was attributed to the early stop of the patient’s recruitment due to technical reasons, but mainly due to the COVID-19 pandemic, where, following strict international recommendations, there was a tendency of a “freeze-all” policy in all IVF Units during the last year. We full acknowledged this limitation, so we proceeded to a post hoc power analysis, which showed a power of 99% for a similar size to our study. Finally, we acknowledge the fact that we included only day three transfers, which is not synchronised with the current growing tendency of “blastocyst” or “freeze-all” policies, but we had to strictly adhere to our initial protocol.

## 5. Conclusions

The use of dexmedetomidine during OR is associated with higher propofol consumption, which does not impair fertilization rates, but is linked with higher quality of embryos transferred, in infertile women undergoing IVF/ICSI. A multicenter RCT, using this study as reference and focusing on live birth rates would replicate or disprove these findings.

## Figures and Tables

**Figure 1 jcm-10-00963-f001:**
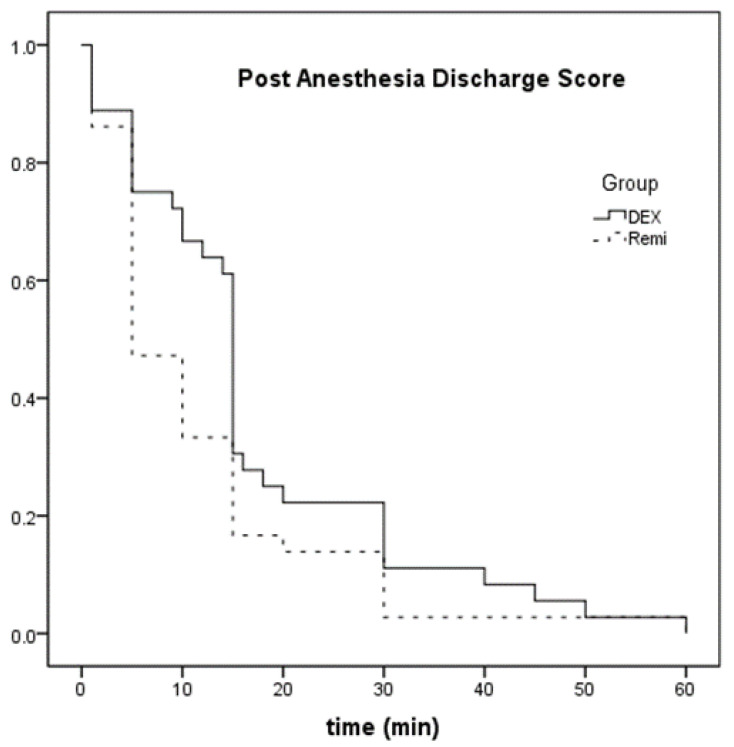
Kaplan Meier analysis for Post Anesthesia Discharge score (see Table 1) between the two groups. It represents the time to event (discharge) in minutes for each of the two groups, according to PADS.

**Table 1 jcm-10-00963-t001:** Scales and scores.

Observer’s Assessment of Alertness/Sedation Scale (OAA/S)		
Responds readily to name spoken in normal tone		5
Lethargic response to name spoken in normal tone		4
Responds only after name is called loudly and/or repeatedly		3
Responds only after mild prodding or shaking		2
Does not respond to mild prodding or shaking		1
**Visual Analogue Scale (VAS)**		
**0____________________________________10**		
No pain	The worst pain	
**Postoperative nausea/vomiting (PONV) assessment**		
**VitalSigns**		2 = within 20% of preoperative value 1 = 20–40% of preoperative value 0 ≥ 40% preoperative value
**Activity and mental status**		2 = Oriented × 3 AND has a steady gait 1 = Oriented × 3 OR has a steady gait 0 = Neither
**Nausea and/or vomiting**		2 = Minimal 1 = Moderate, having required treatment 0 = Severe, requiring treatment
**Pain**		2 = Minimal 1 = Moderate, having required treatment 0 = Severe, requiring treatment
**Surgicalbleeding**		2 = Minimal 1 = Moderate 0 = Severe
none		0
Nausea		1
<2 episodes of vomiting		2
>2 episodes of vomiting		3
**Patients’ and gynecologist’s overall satisfaction assessment**		
Assessment is performed using the following two questions: **1st question:** Are you satisfied with the anesthetic technique used? (Yes/No) **2nd question:** Would you like to use the same anesthetic technique in the future again? (Yes/No)		

**Table 2 jcm-10-00963-t002:** Descriptive statistics. Demographic data of the study population.

	Group DEX (*n* = 36)	Group REMI (*n* = 36)	*p*-Value
Age (years)	38.5 (8.0)	37.5 (5.0)	0.709
BMI (kg/m^2^)	24.3 ± 4.9	24.3 ± 4.1	0.986
Smoking, *n* (%)	9 (25%)	15 (41.6%)	0.211
Alcohol (>4 cups/wk), *n* (%)	5 (13.9%)	4 (11.1%)	0.721
Age of menarche (years)	12.58 ± 1.27	12.83 ±1.13	0.382
AFC	8.13 ± 2.21	8.1 ± 2.64	0.962
AMH (ng/mL)	2.19 ± 0.8	2.16 ± 0.7	0.867
Basal FSH (IU/L)	8.4 ± 1.7	8.1 ± 1.5	0.410
Basal PRL (ng/mL)	9.83 ± 3.3	9.1 ±2.58	0.299
ASA class, *n*			
1	19 (52.7%)	16 (44.4%)	0.638
2	17 (47.2%)	20 (55.5%)	
Infertility, *n*			
Primary	27 (75%)	31 (86%)	0.372
Secondary	9 (25%)	5 (13.8%)	
Cause of infertility, *n* (%)			
Unexplained	17 (47%)	19 (52%)	0.637
Male	7 (19.4%)	6 (16.7%)	0.759
Ovulatory	9 (25%)	6 (16.7%)	0.384
Tubal	3 (8.3%)	5 (13.9%)	0.453
Duration of infertility (years)	2.12 ±0.64	2.25 ± 0.72	0.409
Protocol, *n*			
Long	10 (27.7%)	12 (33.3%)	0.798
Short	26 (72.2%)	24 (66.7%)	
MALE partner			
BMI (kg/m^2^)	25.2 ± 3.6	24.9 ± 3.5	0.720
Smoking (current), *n* (%)	14 (38.9%)	17 (47.2%)	0.475
Alcohol (>4 cups/wk), *n* (%)	9 (25%)	8 (22.2%)	0.781
Varicocele/Cryptorchidism/Obstructions in reproductive tract, *n* (%)	4 (11/1%)	3 (8.3%)	0.69
Previous surgery/Infectious parotitis, *n* (%)	3 (8.3%)	3 (8.3%)	1

BMI: body mass index, ASA: American Society of Anesthesiologists, AFC: antral follicle count, AMH: anti-Müllerian Hormone, FSH: follicle stimulating hormone, PRL: prolactin. Data are presented as mean ± SD, median (IQR) or count (percentage).

**Table 3 jcm-10-00963-t003:** (**a**). Intraoperative and postoperative data. (**b**). In Vitro Fertilization outcome parameters.

(**a**)			
	**Group DEX** **(*n* = 36)**	**Group REMI** **(*n* = 36)**	***p*-Value**
**Intraoperative data**			
Anesthesia duration (min)	22.0 (7.8)	22.0 (7.5)	0.599
Surgery duration (min)	11.5 (7.5)	10.0 (7.8)	0.739
Cumulative Propofol consumption (mg)	81.6 ± 64	17.2 ± 36.4	<0.001
Dexmedetomidine (μg)	27.7 ± 9	-	-
Remifentanil (μg)	-	270 ± 78.5	-
Airway obstruction, *n*	1 (2.7%)	5 (13.9%)	0.199
Need for assisted ventilation, *n*	14 (38.9%)	8 (22.2%)	0.200
Rigidity, *n*	0	0	-
Hypotension, *n*	2 (5.5%)	0	0.493
Bradycardia, *n*	1 (2.7%)	0	1.0
**Postoperative data**			
OAA/S time to 5 (min)	1.18 ± 1.46	1 ± 1.76	0.706
**Patient satisfaction**			
1st question (Yes/No)	31/5	36/0	0.054
2nd question (Yes/No)	35/1	36/0	1.0
**Physician satisfaction**			
1st question (Yes/No)	32/4	36/0	0.120
2nd question (Yes/No)	35/1	36/0	1.0
VAS, 0–10 point	0.27 ± 0.7	0.4 ± 1.15	0.552
PONV, *n*			
Nausea	0	2 (5.5%)	0.2
Vomitus <2 episodes	0	1 (2.7%)	
Hypotension, *n*	0	0	-
Bradycardia, *n*	0	0	-
Post Anesthesia Discharge Score (min)	15.0 (13.5)	5.0 (10.0)	0.028
(**b**)			
	**Group DEX** **(*n* = 36)**	**Group REMI** **(*n* = 36)**	***p*-** **Value**
Number of oocytes retrieved	5 ± 2.3	5.5 ±3.2	0.462
**Embryo quality, *n***			
1	28	18	0.040
2	8	17	
2PN/total number of oocytes	0.6	0.6	0.469
Top quality Day 3	20.7 (5.4)	23.4 (4.7)	0.052
Positive HCG test, *n*	7 (19.4%)	10 (27.7%)	0.580
Clinical pregnancy, *n*	7 (19.4%)	10 (27.7%)	
Miscarriage, *n*	3 (8.3%)	0	0.051

OAA/S: Observer’s Assessment of Alertness/Sedation scale; VAS: Visual Analogue Scale, PONV: postoperative nausea and vomitus. Data are presented as mean ± SD, median (IQR) or count (percentage). Data are presented as mean ± SD, median (IQR) or count (percentage median (IQR) or count (percentage).

## Data Availability

The data presented in this study are available on request from the corresponding author. The data are not publicly available due to reasons of privacy.

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
