# Peer review of "Comparison of Two Different Sedation Protocols during Transvaginal Oocyte Retrieval: Effects on Propofol Consumption and IVF Outcome: A Prospective Cohort Study"

_jcm, 2021, doi:10.3390/jcm10050963_

Round 1
Reviewer 1 Report
I read the manuscript "Comparison of two different sedation protocols during trans-2 vaginal oocyte retrieval: effects on propofol consumption and 3 IVF outcome: a prospective cohort study", with interest.
The study addresses an interesting, often underestimated, topic regarding sedation techniques in the course of oocyte pick-up in IVF cycles. Unfortunately, the sample size of the study is limited as the authors were unable to reach the calculated power due to organizational problems and the COVID-19 pandemic.
Moreover, the baseline data of the couples undergoing IVF and the outcomes of the IVF treatment should be presented in more detail to allow for a good comparison of the results between the two groups.
Detailed Comments
Materials and Methods
Inclusion and exclusion criteria: it is unclear whether patients with endometriosis were included in the study or not. The different distribution of this pathology between the two groups may constitute a selection bias.
Results
Tables 2 and 3b:. So as to interpret the results, it would be useful to clarify the infertility factors that led to IVF, the patients' ovarian reserve at the beginning of the IVF cycle, as well as the ratio between the number of follicles (greater than 15 mm) present at the time of the OR and the number of oocytes retrieved in the two groups.
Moreover, it would be important to provide information on other factors that can affect fertilization rates such as the quality of the semen and the fertilization method (IVF / ICSI) used in the two groups.
In the Methods section, it is stated that embryo freezing was carried out by vitrification. This assumes that some patients had cryopreserved embryo transfers. Therefore, it is important to specify whether the results of pregnancy rates are cumulative (i.e. first fresh and cryopreserved embryo transfers), or not.
Author Response
Suggestions and answers
I read the manuscript "Comparison of two different sedation protocols during trans-2 vaginal oocyte retrieval: effects on propofol consumption and 3 IVF outcome: a prospective cohort study", with interest.
The study addresses an interesting, often underestimated, topic regarding sedation techniques in the course of oocyte pick-up in IVF cycles. Unfortunately, the sample size of the study is limited as the authors were unable to reach the calculated power due to organizational problems and the COVID-19 pandemic.
Moreover, the baseline data of the couples undergoing IVF and the outcomes of the IVF treatment should be presented in more detail to allow for a good comparison of the results between the two groups.
A: We thank the reviewer for the comments. We did our best to improve the quality of this manuscript. Please read the revised text in the points raised, through “track changes” mode.
Detailed Comments
Materials and Methods
Inclusion and exclusion criteria: it is unclear whether patients with endometriosis were included in the study or not. The different distribution of this pathology between the two groups may constitute a selection bias.
A: We agree; due to the limitations of space when describing the study population, we have omitted detailed description; now it reads:”….previous poor ovarian response; ovarian surgery and pathology affecting the endometrial cavity, hereditary or acquired thrombophilia along with surgically proven endometriosis”.
Results
Tables 2 and 3b:. So as to interpret the results, it would be useful to clarify the infertility factors that led to IVF, the patients' ovarian reserve at the beginning of the IVF cycle, as well as the ratio between the number of follicles (greater than 15 mm) present at the time of the OR and the number of oocytes retrieved in the two groups.
A: We agree; we have added in Table 2, the requested parameters. The table now has been substantially upgraded. The number of oocytes is shown in Table 3b.
Moreover, it would be important to provide information on other factors that can affect fertilization rates such as the quality of the semen and the fertilization method (IVF / ICSI) used in the two groups.
A: We have added extra data in Table 2 for the male partner; we have an extra sentence in the methods section (IVF protocol) on ICSI; we had omitted this due to space restrictions. It now reads:”ICSI was performed in all cases to ensure fertilization.”.
In the Methods section, it is stated that embryo freezing was carried out by vitrification. This assumes that some patients had cryopreserved embryo transfers. Therefore, it is important to specify whether the results of pregnancy rates are cumulative (i.e. first fresh and cryopreserved embryo transfers), or not.
Α: We thank the reviewer for the comment. Vitrification is the only cryopreservation method of the particular Unit during the last 12 years. We would be very happy to provide with the cumulative pregnancy rate, but that was not an endpoint of our study and so, we did not manage to get the data.
Reviewer 2 Report
In the manuscript titled “Comparison of two different sedation protocols during transvaginal oocyte retrieval: effects on propofol consumption and IVF outcome: a prospective cohort study" the authors compared the effect of two different regimens for conscious sedation during oocyte Retrieval on total propofol administration, IVF outcome.
- Abstract: It is suggested to change and rewrite the first sentence of the background.
- Page 7, line 218 please delete “all” in the parentheses.
- It is suggested to add the formula for Sample size.
- For figure 1, please determine what is the vertical line and explain its value.
- On page 8, line 238, please clarify the “the rest of the secondary outcomes”.
- Please add the time of the study (from x to y) to the method.
- Discussion is lacking depth and purely descriptive. Please rewrite it.
- Some other lifestyle and socioeconomic factors can affect the results of IVF. They were not matched in two groups of the study.
- The quality of sperm and the age of the man is another affecting factor. There is not any information from the other partner in the manuscript.
Author Response
In the manuscript titled “Comparison of two different sedation protocols during transvaginal oocyte retrieval: effects on propofol consumption and IVF outcome: a prospective cohort study" the authors compared the effect of two different regimens for conscious sedation during oocyte Retrieval on total propofol administration, IVF outcome.
- Abstract: It is suggested to change and rewrite the first sentence of the background.
A: We have rewritten the first sentence, as suggested. It now reads:”Background: There has been various reports on the potential impact of anesthetic agents used during oocyte retrieval (OR)on the impairment of the capacity of the oocyte for fertilization and subsequent embryo quality; results have been conflicting;”.
- Page 7, line 218 please delete “all” in the parentheses.
A: Deleted as suggested.
- It is suggested to add the formula for Sample size.
A: We have changed the reporting in the respective part; It now reads:“Using the G*Power 3.0.10 software, we found that a sample size of 86 patients in each group would be necessary to detect a 30% dose difference in propofol using Fisher's exact double-sided alpha test at the 0.05 level, including a 5% withdrawal rate from the study (Appendix).”. We have also added extra data on this, in the Appendix (formula and results) to comply with the suggestion.
- For figure 1, please determine what is the vertical line and explain its value.
A: We have added an extra sentence in Figure; it now reads: “Kaplan Meier analysis for Post Anesthesia Discharge score (see Appendix) between the two groups. It represents the time to event (discharge) in minutes for each of the two groups, according to PADS.”.
- On page 8, line 238, please clarify the “the rest of the secondary outcomes”.
A: we thank the reviewer for the comment. We have amended the test; it now reads:”As for the secondary outcomes preset for this study, only the quality of embryos three days post OR were higher, while discharge scores were lower in the study in comparison to the control group.”.
- Please add the time of the study (fromx to y) to the method.
A: we thank the reviewer for the comment. We have amended the test; it now reads:”The study was performed in accordance with good clinical practice guidelines and the ethical standards of the Helsinki Declaration, lasted from November 2017 to November 2019, while written informed consent was obtained from all women.”.
Discussion is lacking depth and purely descriptive. Please rewrite it.
A: We thank the reviewer for the comment. We have amended the discussion section accordingly; we have added 2 new references and 2 new paragraphs to comply with the suggestion. Please check the text through the “track changes” mode.
- Some other lifestyle and socioeconomic factors can affect the results of IVF. They were not matched in two groups of the study.
A: We agree; we have added in Table 2, the requested parameters. The table now has been substantially upgraded.Please check the text through the “track changes” mode.
- The quality of sperm and the age of the man is another affecting factor. There is not any information from the other partner in the manuscript.
A: We agree; we have added in Table 2, the requested parameters (we have added a separate section for the male partner). The table now has been substantially upgraded.
Reviewer 3 Report
The authors present a prospective controlled trial to evaluate effects of anesthetic agents on oocyte function and embryonic capacity.
English errors throughout the manuscript need to be addressed.
The term, "normo-ovulatory" is not scientific, and appears to exclude many women who would require IVF for fertility treatment (if you are normally ovulating, you may not require IVF treatment.)
Details of patient assignment should be provided more clearly provided. As stated, the paper suggests that patients were sequentially assigned based on case order. This is not an ideal randomization process.
Day 3 transfer is not typical for many IVF centres at this time, which represents a limitation of the study.
Clinical pregnancy is usually defined as the presence of an intrauterine sac with fetal heart beat.
Since only 36 of a projected 86 couples were enrolled in each group, the ability of the study to project "equivalence" should be reconsidered. An additional power calculation with 36 cycles per intervention group should be provided.
The authors should further discuss the improvement in embryo quality seen in the study, as well as the potential risks of increased propafol use.
Author Response
The authors present a prospective controlled trial to evaluate effects of anesthetic agents on oocyte function and embryonic capacity.
English errors throughout the manuscript need to be addressed.
The term, "normo-ovulatory" is not scientific, and appears to exclude many women who would require IVF for fertility treatment (if you are normally ovulating, you may not require IVF treatment.)
A: We agree; we have amended the definition accordingly; it now reads:“….age 25-43 years; cycle length between 22 and 35 days; and basal FSH ≤11 mIU/mL.”.
Details of patient assignment should be provided more clearly provided. As stated, the paper suggests that patients were sequentially assigned based on case order. This is not an ideal randomization process.
A: we thank the reviewer for the comment. As our study is not an RCT, but a cohort, we did not proceed to a proper randomization procedure. We state that the allocation to each group was done according to the random order that participants came to the Unit; we fully acknowledge that this inserts selection bias unavoidably, and we specifically state this limitation in the limitations section of our study. We have written:”Limitations of the study include its non-randomized design that is linked with known and unknown confounding and biases [28]”.
Day 3 transfer is not typical for many IVF centres at this time, which represents a limitation of the study.
A: We thank the reviewer for the comment. As the day of transfer and the relevant policies (for example “freeze all” policy and “blastocyst policy”) are changing throughout time, our protocol included only d3 transfers and we followed it strictly, as we had to. We specifically state this limitation in the limitations section of our study. We have added the sentence:”Finally, we acknowledge the fact that we included only day three transfers, which is not synchronized with the current growing tendency of “blastocyst” or “freeze-all” policies, but we had to strictly adhere to our initial protocol.”.
Clinical pregnancy is usually defined as the presence of an intrauterine sac with fetal heart beat.
A: we agree; we have amended the definition accordingly; it now reads:“defined as the presence of heart beat of the embryo at 7 weeks gestation”.
Since only 36 of a projected 86 couples were enrolled in each group, the ability of the study to project "equivalence" should be reconsidered. An additional power calculation with 36 cycles per intervention group should be provided.
A: we thank the reviewer for the comment. We totally agree. As allocation was done according to a principle of 1:1, it was logical that we would have equal numbers. Secondly, post hoc power analysis of the data on propofol consumption was also carried out to check the power of the study results, in an effort to reduce the bias associated with the premature ending of the study. This is explicitly stated in the second paragraph of the sample size section. It reads: “A post hoc power analysis of the data on propofol consumption was also carried out to check the power of the study results, in an effort to reduce the bias associated with the premature ending of the study.“.
The authors should further discuss the improvement in embryo quality seen in the study, as well as the potential risks of increased propafol use.
A: We thank the reviewer for the comment. We have amended the discussion section accordingly; we have added 2 new references and 2 new paragraphs to comply with the suggestions. Please check the text through the “track changes” mode.